# PROVABLE REPAIR OF VISION TRANSFORMERS: LAST LAYER IS ALL YOU NEED

## ABSTRACT

Vision Transformers have emerged as state-of-the-art image recognition tools, but may still exhibit incorrect behavior. Incorrect image recognition can have disastrous consequences in safety-critical real-world applications such as self-driving automobiles. In this paper, we present Provable Repair of Vision Transformers (PRoViT), a provable repair approach that guarantees the correct classification of images in a repair set for a given Vision Transformer without modifying its architecture. PRoViT avoids negatively affecting correctly classified images (drawdown) by minimizing the changes made to the Vision Transformer's parameters and original output. We observe that for Vision Transformers, unlike for other architectures such as ResNet or VGG, editing just the parameters in the last layer achieves correctness guarantees and very low drawdown. We introduce a novel method for editing these last-layer parameters that enables PRoViT to efficiently repair state-of-the-art Vision Transformers for thousands of images, far exceeding the capabilities of prior provable repair approaches.

## 1 INTRODUCTION

Vision Transformers (Dosovitskiy et al., 2021) have emerged as state-of-the-art image recognition tools, but still exhibit faulty behavior that can result in disastrous real-world consequences. Image recognition plays a significant role in safety-critical applications such as self-driving automobiles (Bojarski et al., 2016) and medical diagnosis (Kermany et al., 2018). Faulty image recognition software has resulted in serious consequences, including loss of life (Gonzales, 2019; Lee, 2016). As Vision Transformers integrate more into real-world applications, it becomes increasingly important to provide guarantees about their correctness to ensure safety.

There is an abundance of research on editing pre-trained Transformers to correct faulty behavior (Meng et al., 2022; 2023; Mitchell et al., 2022; Huang et al., 2023). To the best of our knowledge, none of these methods provide provable correctness guarantees. Recent research on provable repair of deep neural networks (DNNs) (Sotoudeh & Thakur, 2021; Tao et al., 2023; Goldberger et al., 2020; Fu & Li, 2022) explores strategies to provide such guarantees, but such research has not focused on Vision Transformers. In general, provable repair methods guarantee correctness of a DNN's output according to a user-defined repair specification. Provable repair methods strive for the following properties:

- *Efficacy*: The repaired DNN must achieve 100% accuracy on the specified points.
- *Efficiency*: The repair process should be efficient and scale to large DNNs.
- *Low drawdown*: The repair should not negatively affect the previous good behavior of a DNN.
- *High generalization*: The repair should generalize to similar points that are not directly specified in the repair set.

These properties set provable repair apart from other DNN editing methods such as retraining and fine tuning. Retraining is inefficient, especially on large models, and the original training set may not be available. Fine tuning has demonstrated a tendency to cause high drawdown (Kemker et al., 2018), meaning the edited DNN has "forgotten" much of its original knowledge.

In this paper, we present PRoViT, a provable repair method for Vision Transformers that provides correctness guarantees without modifying the original model's architecture. PRoViT is *sound*: a repaired network returned by PRoViT is guaranteed to classify all points in the repair set correctly. Similar to prior provable repair approaches (Goldberger et al., 2020; Sotoudeh & Thakur, 2021; Tao et al., 2023), PRoViT is *not complete*: given a network and a repair specification, it may not find a repaired network. In practice, however, PRoViT successfully repairs thousands of images on state-of-the-art Vision Transformers. It is efficient and highly scalable, and it avoids drawdown by minimizing the changes made to the parameters of the Vision Transformer.

The key observation underlying PRoViT is that editing the parameters of the last fully-connected linear layer of the Vision Transformer is sufficient to achieve provable repair with low drawdown and high generalization. In particular, PRoViT combines gradient-descent-based fine tuning of the last layer with a scalable linear-programming-based provable repair approach to edit the parameters of the last layer. PRoViT is able to repair ImageNet Vision Transformers using thousands of repair points, far exceeding prior approaches for provable repair such as APRNN (Tao et al., 2023) and "minimal modifications of deep neural networks" (MMDNN) (Goldberger et al., 2020). Though APRNN and MMDNN present strategies to repair the last layer of a DNN by formulating the repair as a linear programming (LP) problem, they are unable to scale beyond a few hundred repair points. This is because these earlier LP formulations require a large number of variables and constraints, which is avoided by the novel approach presented in this paper.

To the best of our knowledge, PRoViT is the only approach for provable repair of Vision Transformers with all of the following properties:

- *Provable correctness guarantees*: The repaired network returned by PRoViT is guaranteed to classify all points in the repair set.
- *Transformer architecture-preserving*: The repair does not make any changes to the original architecture of the Vision Transformer.
- *Highly scalable*: PRoViT successfully repairs large Vision Transformers and repair sets with thousands of images.
- *Efficient*: The repair is efficient, remaining within the order of minutes to hours for thousands of points.
- *Low drawdown*: PRoViT does not negatively impact the previous correct classifications of the Vision Transformer after repairing the images in the repair set.
- *High generalization*: The repair generalizes to images beyond those explicitly present in the repair set.

The rest of the paper is organized as follows: Section 2 presents preliminaries; Section 3 presents the PRoViT approach; Section 4 details the experimental evaluation of PRoViT; Section 5 discusses related work; Section 6 concludes.

## 2 PRELIMINARIES

We use $\mathcal{N}^\theta$ to denote a deep neural network (DNN) with parameters $\theta$, and $\mathcal{N}(x; \theta) \in \mathbb{R}^n$ to denote the output vector of the DNN on input vector $x \in \mathbb{R}^m$. We drop the parameters $\theta$ if they are clear from the context. In this paper, we restrict ourselves to classification tasks; thus, $n$ is the number of labels. We use $accuracy(\mathcal{N}^\theta, \Lambda)$ to represent the accuracy of a DNN $\mathcal{N}^\theta$ on set $\Lambda$ of inputs and expected labels.

Given a *repair set* $\mathcal{S}$ of inputs and expected labels, the goal of *architecture-preserving provable repair* is to make small changes to the parameters of a given DNN $\mathcal{N}^\theta$ so that the resulting DNN $\mathcal{N}^{\theta'}$ has 100% accuracy on the repair set $\mathcal{S}$.

**Definition 1.** *Given a DNN $\mathcal{N}^\theta$ and a* repair set $\mathcal{S}$ *of inputs and expected labels, an* architecture-preserving provable repair *finds parameters $\theta'$ such that $\bigwedge_{(x,l) \in \mathcal{S}} \arg\max(\mathcal{N}(x; \theta')) = l$; that is,* $accuracy(\mathcal{N}^{\theta'}, \mathcal{S}) = 100\%.$

We use *efficacy* to refer to the accuracy of the repaired network on the given repair set. Apart from efficacy, provable repair methods are also evaluated on *drawdown* and *generalization*.

**Definition 2.** *A drawdown set $\mathcal{D}$ is a set of points disjoint from the repair set and representative of a DNN's existing knowledge. For two DNNs $\mathcal{N}$ and $\mathcal{N}'$, the drawdown of $\mathcal{N}'$ with respect to $\mathcal{N}$ is $accuracy(\mathcal{N}, \mathcal{D}) - accuracy(\mathcal{N}', \mathcal{D})$. Lower drawdown is better, representing less knowledge lost during repair.*

**Definition 3.** *A generalization set $\mathcal{G}$ is a set of points disjoint but similar to those in the repair set. For two DNNs $\mathcal{N}$ and $\mathcal{N}'$, the generalization of $\mathcal{N}'$ with respect to $\mathcal{N}$ is $accuracy(\mathcal{N}', \mathcal{G}) - accuracy(\mathcal{N}, \mathcal{G})$. Higher generalization is better.*

Our goal is to find an architecture-preserving provable repair approach for Vision Transformers (Dosovitskiy et al., 2021). Vision Transformers are self-attention-based architectures that partition an input image into patches for processing. Encoder layers process the patches, taking into account their relations to one another via nonlinear operations. Finally, the last encoder layer returns a class token, which is a vector representing the predicted class of the input image. This class token is passed through the final feed-forward layer of the Vision Transformer to determine the label for the input image.

There are a number of existing provable repair approaches including PRDNN (Sotoudeh & Thakur, 2021), REASSURE (Fu & Li, 2022), MMDNN (Goldberger et al., 2020), and APRNN (Tao et al., 2023). PRDNN is unable to repair Vision Transformers due to the nonlinearity and architecture within the encoder layers. REASSURE only works for models that use ReLU activation functions, but Transformers use softmax activations. In addition, PRDNN and REASSURE are not architecture-preserving, so we turn to MMDNN and APRNN instead.

MMDNN and APRNN encounter similar issues in the encoder layers as REASSURE and PRDNN, but MMDNN and APRNN can both narrow their focus to only modify the last layer of a model and encode the repair as a linear programming (LP) problem.

We use $\mathcal{N}^{(:-1)}$ to represent a DNN $\mathcal{N}$ without its last layer $\mathcal{N}^{(-1)}$. Similarly, we use $\theta^{(:-1)}$ to denote $\mathcal{N}$'s parameters excluding those in the last layer $\theta^{(-1)}$. Thus, $\mathcal{N}^{(:-1)}(\boldsymbol{x}; \theta^{(:-1)}) \in \mathbb{R}^p$ is the input vector to the last layer of $\mathcal{N}$ for some input vector $\boldsymbol{x}$. The parameters $\theta^{(-1)} \stackrel{\text{def}}{=} \{\boldsymbol{W}, \boldsymbol{b}\}$ of the last layer consist of weights $\boldsymbol{W}$ and biases $\boldsymbol{b}$. $\boldsymbol{W}$ has shape $p \times n$ and $\boldsymbol{b}$ has shape $n$, where $n$ is the number of labels in the classification task.

First, we introduce symbolic parameters $\widehat{\theta}^{(-1)} \stackrel{\text{def}}{=} \{\widehat{\boldsymbol{W}}, \widehat{\boldsymbol{b}}\}$ where $\widehat{\boldsymbol{W}}$ is a symbolic matrix corresponds to $\boldsymbol{W}$ and $\widehat{\boldsymbol{b}}$ is a symbolic bias vector corresponds to $\boldsymbol{b}$. $\widehat{\boldsymbol{W}}$ and $\widehat{\boldsymbol{b}}$ represent the new weights and biases we must find to satisfy the repair set. Now, the output of the layer is $\widehat{\boldsymbol{y}} = \mathcal{N}^{(:-1)}(\boldsymbol{x}; \theta^{(:-1)}) \cdot \widehat{\boldsymbol{W}} + \widehat{\boldsymbol{b}}$ for an input vector $\boldsymbol{x}$. $\widehat{\boldsymbol{y}}$ is a symbolic output vector with $n$ elements, one for each label.

The following formula $\varphi_{\text{argmax}}(\widehat{\boldsymbol{y}}, l)$ means the argmax of the vector $\widehat{\boldsymbol{y}}$ is $l$:

$$\varphi_{\text{argmax}}(\widehat{\boldsymbol{y}}, l) \stackrel{\text{def}}{=} \bigwedge_{i \neq l} \widehat{\boldsymbol{y}}_l > \widehat{\boldsymbol{y}}_i \tag{1}$$

Intuitively, representing the argmax function in the form of linear constraints requires that the value of $\widehat{\boldsymbol{y}}_l$ is greater than all other values $\widehat{\boldsymbol{y}}_i$ in $\widehat{\boldsymbol{y}}$ where $i \neq l$. Using this $\varphi_{\text{argmax}}$ definition, we formulate the LP for provable repair as follows:

$$\min \left\| \boldsymbol{W} - \widehat{\boldsymbol{W}} \right\| + \left\| \boldsymbol{b} - \widehat{\boldsymbol{b}} \right\|$$
$$\text{s.t.} \bigwedge_{(\boldsymbol{x}, l) \in \mathcal{S}} \left( \varphi_{\text{argmax}}(\widehat{\boldsymbol{y}}, l) \ \wedge \ \widehat{\boldsymbol{y}} = \mathcal{N}^{(:-1)}(\boldsymbol{x}; \theta^{(:-1)}) \cdot \widehat{\boldsymbol{W}} + \widehat{\boldsymbol{b}} \ \wedge \ \widehat{\boldsymbol{y}}_l > \max(\mathcal{N}(\boldsymbol{x}, \theta)) \right) \tag{2}$$

An off-the-shelf LP solver can find a solution to $\widehat{\boldsymbol{W}}$ and $\widehat{\boldsymbol{b}}$ that satisfies all of the constraints if it exists. We then update the original parameters of the last layer, $\theta^{(-1)}$, with the solution to make $\theta^{(-1)}_{new}$. The repaired $\mathcal{N}$'s parameters are $\theta' = [\theta^{(:-1)}, \theta^{(-1)}_{new}]$.

**Proposition 1** (Goldberger et al. (2020)). *Given a repair set $\mathcal{S}$ and DNN $\mathcal{N}$, $\theta'$ solved by the LP in Equation 2 satisfies $\mathcal{S}$.*

**Remark 1.** *The number of variables in the LP in Equation 2 is $n \times p + n$ and the number of constraints is $(n - 1) \times |\mathcal{S}|$.*

*Proof.* The symbolic matrix $\widehat{W}$ is of size $p \times n$, and each element of $\widehat{W}$ is a variable. Similarly, the symbolic vector $\widehat{b}$ is of size $n$, and each element of $\widehat{b}$ is a variable. In total, there are $n \times p + n$ variables in the LP.

For each element in the repair set, we add $n - 1$ constraints. Each of these constraints ensures that the symbolic output associated with the correct label is greater than another label. Because there are $n$ labels, there must be $n - 1$ comparison constraints (as there is no need to add a constraint to compare the correct label with itself). There are $|\mathcal{S}|$ points in the repair set, so we multiply $|\mathcal{S}|$ with $n - 1$ to get $(n - 1) \times |\mathcal{S}|$ total constraints. □

The LP for provable repair in Equation 2 is categorized as an architecture-preserving provable repair approach for Vision Transformers. Consider another approach in this category based on fine-tuning. Fine tuning is a well-studied strategy to adjust a DNN's behavior on a set of points, usually disjoint from the training set. The parameters $\theta$ of a DNN $\mathcal{N}$ are updated via gradient descent. We define a variation of fine tuning, $\text{FT}_{\text{all}}$, that continues to edit all of the parameters of the DNN until the repair set is satisfied. $\text{FT}_{\text{all}}$ is not guaranteed to terminate, but if it terminates, then all inputs are classified correctly and hence it is a provable repair approach. We will consider $\text{FT}_{\text{all}}$ a baseline to compare against our approach.

## 3 APPROACH

In this section, we present our scalable architecture-preserving provable repair method for Vision Transformers, PRoViT. A key observation in this paper is that editing the parameters of the last layer of a Vision Transformer is sufficient to find a high-quality provable repair, namely, one with low drawdown and high generalization. We present three variants of PRoViT:

- **PRoViT$_{\text{LP}}$**: PRoViT$_{\text{LP}}$ is a novel scalable LP repair approach, the details of which are provided later in this section. We observed that the LP for provable repair defined in Equation 2 works well on small repair sets, resulting in very low drawdown and high generalization, but it does not scale beyond 250 images. The LP solver runs out of memory due to the high number of variables and constraints required to encode the repair. Instead, our reduced LP approach PRoViT$_{\text{LP}}$ results in very low drawdown, generalizes well, and scales to thousands of images, as shown in Section 4.

- **PRoViT$_{\text{FT}}$**: PRoViT$_{\text{FT}}$ is a variation on fine tuning which restricts the edits to just the last layer of the Vision Transformer. We observed that $\text{FT}_{\text{all}}$ results in high drawdown and is inredibly inefficient. On the other hand, PRoViT$_{\text{FT}}$ is much more efficient and has lower drawdown than $\text{FT}_{\text{all}}$, as shown in Section 4.

- **PRoViT$_{\text{FT+LP}}$**: PRoViT$_{\text{FT+LP}}$ is a combination of the two above approaches. PRoViT$_{\text{LP}}$ results in extremely low drawdown and generalizes well. PRoViT$_{\text{FT}}$ generalizes better than PRoViT$_{\text{LP}}$ but has slightly higher drawdown. PRoViT$_{\text{FT+LP}}$ leverages the strengths of both to find a satisfying tradeoff between drawdown and generalization. We demonstrate this tradeoff in Section 4.

Now we explain our scalable LP provable repair approach, PRoViT$_{\text{LP}}$. Let $(\boldsymbol{x}, l)$ be an element of the repair set. It is sufficient to only modify the value of $\boldsymbol{y}_l$ to repair the DNN for $\boldsymbol{x}$. Consequently, it is sufficient to only modify the $l$-th column of the last-layer weight matrix $W$. Now let $K$ be the set of labels present in the entire repair set. Our new LP formulation PRoViT$_{\text{LP}}$ is based on the observation that it is sufficient *in practice* to only adjust the values of *those particular* $|K|$ elements of $\boldsymbol{y}$ and not modify the rest. Consequently, it is sufficient in practice to only modify the columns in $K$ of the last-layer weight and bias matrices.

Let us now convert $K$ to a subsequence of $[0, 1, \ldots, n - 1]$ where $n$ is the total number of labels in the classification task. We define a submatrix of $\boldsymbol{W}$, denoted $\boldsymbol{W}_{:,K}$, and a subvector of $\boldsymbol{b}$, denoted $\boldsymbol{b}_K$. These submatrices are formed by selecting columns of $\boldsymbol{W}$ and $\boldsymbol{b}$ indexed by $K$. $\boldsymbol{W}_{:,K}$ has shape $p \times |K|$ and $\boldsymbol{b}_K$ has shape $|K|$ where $p$ is the size of the input to the last layer $\mathcal{N}^{(-1)}$.

**Example 1.** *Consider a matrix* $\boldsymbol{W} = \begin{bmatrix} 1 & 3 & 5 & 7 & 9 \\ 2 & 4 & 6 & 8 & 10 \end{bmatrix}$. *Then* $\boldsymbol{W}_{:,Q} = \begin{bmatrix} 3 & 5 & 9 \\ 4 & 6 & 10 \end{bmatrix}$ *where* $Q = [1, 2, 4]$.

---

**Algorithm 1:** PRoViT$_{\text{FT+LP}}(\mathcal{N}, \mathcal{S})$

---

**Input:** A Vision Transformer $\mathcal{N}$ and a repair set $\mathcal{S}$.
**Output:** A repaired Vision Transformer $\mathcal{N}'$ that satisfies $\mathcal{S}$.
1 PRoViT$_{\text{FT}}(\mathcal{N}, \mathcal{S})$
2 **while** $\neg$PRoViT$_{\text{LP}}(\mathcal{N}, \mathcal{S})$ **do**
3 $\quad\lfloor$ PRoViT$_{\text{FT}}(\mathcal{N}, \mathcal{S})$
4 **return** $\mathcal{N}$

---

We introduce a symbolic matrix $\widehat{\boldsymbol{W}}_{reduced}$ and a symbolic vector $\widehat{\boldsymbol{b}}_{reduced}$ to represent the weights and biases we must find to satisfy the repair set $\mathcal{S}$. The shapes of $\widehat{\boldsymbol{W}}_{reduced}$ and $\widehat{\boldsymbol{b}}_{reduced}$ match $\boldsymbol{W}_{:,K}$ and $\boldsymbol{b}_K$, respectively, since we will only find new values for the weights and biases associated with the labels in $K$.

We encode the constraints to ensure that each repair point is correctly classified. Let $\max(Y)$ be a function that returns the maximum value in the vector $Y$. We formulate the reduced LP for PRoViT as follows:

$$\min \left\| \boldsymbol{W}_{:,K} - \widehat{\boldsymbol{W}}_{reduced} \right\| + \left\| \boldsymbol{b}_K - \widehat{\boldsymbol{b}}_{reduced} \right\|$$

$$\text{s.t.} \bigwedge_{(\boldsymbol{x},l) \in \mathcal{S}} \left( \varphi_{\text{argmax}}(\widehat{\boldsymbol{y}}, l) \ \wedge \ \widehat{\boldsymbol{y}} = \mathcal{N}^{(:-1)}(\boldsymbol{x}; \theta^{(:-1)}) \cdot \widehat{\boldsymbol{W}}_{reduced} + \widehat{\boldsymbol{b}}_{reduced} \ \wedge \ \widehat{\boldsymbol{y}}_l > \max(\mathcal{N}(\boldsymbol{x}, \theta)) \right)$$

(3)

An LP solver can find a solution to $\widehat{\boldsymbol{W}}_{reduced}$ and $\widehat{\boldsymbol{b}}_{reduced}$ that satisfies all of the constraints if it exists. We then update the original parameters of the last layer, $\theta^{(-1)}$, with the solution to make $\theta_{new}^{(-1)}$. The repaired $\mathcal{N}$'s parameters are $\theta' = [\theta^{(:-1)}, \theta_{new}^{(-1)}]$.

**Proposition 2.** *Given a repair set $\mathcal{S}$ and DNN $\mathcal{N}$ with parameters $\theta$, the repaired DNN with parameters $\theta'$ solved by the LP in Equation 3 satisfies $\mathcal{S}$.*

*Proof.* Let $\theta'$ be the parameters computed by the provable repair technique that uses Equation 3. Let $(\boldsymbol{x}, l)$ be any element of the repair set $\mathcal{S}$, and $\boldsymbol{y}' = \mathcal{N}(\boldsymbol{x}, \theta')$. We will show that $\arg\max(\boldsymbol{y}') = l$.

The argmax constraints $\varphi_{\text{argmax}}$ ensure that

$$\boldsymbol{y}'_l > \boldsymbol{y}'_i \text{ for all } i \in K - \{l\} \tag{4}$$

Let $\boldsymbol{y} = \mathcal{N}(\boldsymbol{x}, \theta)$. Equation 3 includes a constraint to ensure that $\boldsymbol{y}'_l > \boldsymbol{y}_i$ for all $i \in |\boldsymbol{y}|$. The outputs not associated with the labels in $K$ are not modified by the repair; thus, $\boldsymbol{y}'_i = \boldsymbol{y}_i$ for all $i \notin K$.

$$\boldsymbol{y}'_l > \boldsymbol{y}'_i \text{ for all } i \notin K \tag{5}$$

Using Equations 4 and 5, we have $\boldsymbol{y}'_l > \boldsymbol{y}'_i, i \neq l$; that is, $\arg\max(\boldsymbol{y}') = l$. $\qquad\square$

**Remark 2.** *The number of variables in the LP in Equation 3 is $p \times |K| + |K|$ and the number of constraints is $|\mathcal{S}| \times |K|$.*

*Proof.* The size of the symbolic matrix $\widehat{\boldsymbol{W}}_{reduced}$ is $|K| \times p$ and each element of $\widehat{\boldsymbol{W}}_{reduced}$ is a variable. Similarly, the symbolic vector $\widehat{\boldsymbol{b}}_{reduced}$ is of size $|K|$ and each element of $\widehat{\boldsymbol{b}}_{reduced}$ is a variable. In total, there are $p \times |K| + |K|$ variables in the LP.

The $|K|$ constraints consist of $|K| - 1$ constraints to encode the argmax across the labels present in $K$. There is one more constraint added to encode the max function. There are $|K| - 1 + 1 = |K|$ constraints per element in the repair set, so there are $|K| \times |\mathcal{S}|$. $\qquad\square$

As demonstrate in Remark 1 and Remark 2, while the size of the original problem (Equation 2) depends on the number of labels $n$, the size of the reduced problem (Equation 3) depends on $|K| \leq n$.

PRoViT$_{\text{FT+LP}}$ combines both PRoViT$_{\text{FT}}$ and PRoViT$_{\text{LP}}$ as shown in Algorithm 1. Given a Vision Transformer $\mathcal{N}$ and a repair set $\mathcal{S}$, PRoViT$_{\text{FT+LP}}$ first runs one iteration of PRoViT$_{\text{FT}}$ to quickly gain accuracy of $\mathcal{N}$ on the inputs in $\mathcal{S}$. PRoViT$_{\text{FT}}$ may achieve 100% efficacy at this stage, in which case PRoViT$_{\text{FT+LP}}$ terminates and returns the repaired $\mathcal{N}$. If the efficacy is not 100%, PRoViT$_{\text{FT+LP}}$ runs PRoViT$_{\text{LP}}$ to make additional edits to ensure that all inputs in $\mathcal{S}$ are classified correctly. If there is no solution to the LP, the loop continues. Otherwise, the repaired Vision Transformer is returned.

Table 1: Drawdown and generalization results in the experiment. **Bold number** indicates the *best result*, underlined number indicates the *second best result*, t/o indicates timeout in 20000 seconds.

| Model | $|K|$ | $|\mathcal{S}|$ | Drawdown [%] | | | | Generalization [%] | | | |
|-------|-------|-----------------|--------------|--------|--------|--------|--------------------|--------|--------|--------|
| | | | FT$_{all}$ | PRoViT | | | FT$_{all}$ | PRoViT | | |
| | | | | LP | FT | FT+LP | | LP | FT | FT+LP |
| | 4 | 2000 | 76.80% | **0.01%** | 0.22% | 0.08% | 24.70% | 43.82% | **53.40%** | 49.07% |
| | 8 | 4000 | 76.77% | **0.01%** | 0.67% | 0.23% | 8.31% | 32.18% | **39.74%** | 38.96% |
| ViT-L/32 | 12 | 6000 | 76.66% | **0.01%** | 1.05% | 0.39% | 8.35% | 35.25% | **42.14%** | 41.65% |
| | 16 | 8000 | t/o | **0.01%** | 2.19% | 0.57% | t/o | 32.43% | 37.80% | **38.16%** |
| | 20 | 10000 | t/o | **0.02%** | 3.24% | 0.74% | t/o | 31.69% | 36.20% | **37.76%** |
| | 4 | 2000 | 81.51% | **-0.01%** | 0.28% | **-0.01%** | 36.18% | 44.76% | **54.33%** | 50.01% |
| | 8 | 4000 | 81.39% | **-0.01%** | 0.80% | 0.06% | 17.21% | 32.87% | **39.34%** | 38.05% |
| DeiT | 12 | 6000 | t/o | **0.00%** | 1.43% | 0.14% | t/o | 34.77% | **41.74%** | 40.37% |
| | 16 | 8000 | t/o | **0.05%** | 2.40% | 0.40% | t/o | 31.99% | 37.41% | **38.02%** |
| | 20 | 10000 | t/o | **0.05%** | 4.33% | 0.62% | t/o | 31.45% | 35.75% | **37.49%** |

Table 2: Time spent in the experiment. **Bold number** indicates the *best result*, underlined number indicates the *second best result*, t/o indicates timeout in 20000 seconds.

| Model | $|K|$ | $|\mathcal{S}|$ | Time [s] | | | |
|-------|-------|-----------------|----------|--------|--------|--------|
| | | | FT$_{all}$ | PRoViT | | |
| | | | | LP | FT | FT+LP |
| | 4 | 2000 | 8502s | **440s** | 537s | 541s |
| | 8 | 4000 | 12633s | 964s | **847s** | 1187s |
| ViT-L/32 | 12 | 6000 | 18869s | 1598s | **1230s** | 1834s |
| | 16 | 8000 | t/o | **2339s** | 3225s | 2637s |
| | 20 | 10000 | t/o | **3095s** | 3960s | 3576s |
| | 4 | 2000 | 2681s | **386s** | 2282s | 535s |
| | 8 | 4000 | 9853s | **820s** | 1950s | 1204s |
| DeiT | 12 | 6000 | t/o | **1317s** | 1944s | 1761s |
| | 16 | 8000 | t/o | **1952s** | 2642s | 2473s |
| | 20 | 10000 | t/o | **2576s** | 3050s | 3291s |

## 4 EXPERIMENTAL EVALUATION

The following experiments repair three Vision Transformers trained on ImageNet: ViT-L/32 (Dosovitskiy et al., 2021) and DeiT (Touvron et al., 2021). We also evaluate our approach on ResNet152 (He et al., 2016) and VGG19 (Simonyan & Zisserman, 2015) to demonstrate that last layer repairs are best suited for Vision Transformers rather than other image recognition architectures. All experiments were run on a machine with dual 16-core Intel Xeon Silver 4216 CPUs, 384 GB of memory, SSD and a NVIDIA RTX A6000 with 48 GB of GPU memory. We implemented PRoViT using PyTorch (Paszke et al., 2019) and Gurobi (Gurobi Optimization, LLC, 2023), a mathematical optimization solver for linear programming (LP) problems.

***Baseline and our approach.*** Our baseline is **FT$_{all}$**. We could not evaluate full Vision Transformers on a standard LP-based last-layer repair (as in Equation 2) due to the scalability issues discussed in Section 3. However, we compared the performance of APRNN (Tao et al., 2023) on reduced Vision

Table 3: Drawdown, generalization and timing results for non-ViT networks in the experiment. **Bold number** indicates the *best result*, underlined number indicates the *second best result*, t/o indicates timeout in 20000 seconds.

| Model | $|K|$ | $|\mathcal{S}|$ | Drawdown [%] PRoViT | | | Generalization [%] PRoViT | | | Time [s] PRoViT | | |
|---|---|---|---|---|---|---|---|---|---|---|---|
| | | | LP | FT | FT+LP | LP | FT | FT+LP | LP | FT | FT+LP |
| ResNet152 | 4 | 2000 | **11.28%** | 77.92% | 77.87% | 52.90% | **56.58%** | 55.94% | 1851s | **1323s** | 2004s |
| VGG19 | 4 | 2000 | **1.03%** | 56.01% | 45.97% | 50.14% | **52.92%** | 52.68% | **621s** | 4784s | 848s |

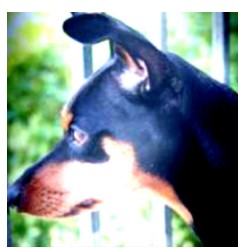
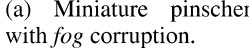
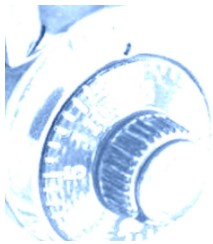
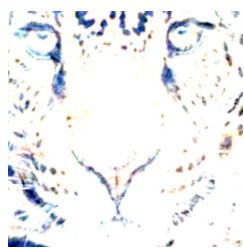
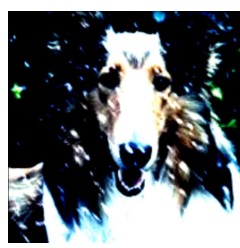

(a) Miniature pinscher with *fog* corruption.

(b) Combination lock with *brightness* corruption.

(c) Jaguar with *frost* corruption.

(d) Collie with *snow* corruption.

Figure 1: Examples of images with different corruptions applied

Transformers with just 50 ImageNet classes and noted that PRoViT outperforms APRNN even in the reduced scenario. The results of this comparison are in the Appendix. Other baselines were eliminated because they do not support transformer architectures or their approaches were specific to NLP tasks. We evaluate the following three variants of our approach PRoViT: (1) **PRoViT_LP** (abbrev. **LP**): without any initial fine tuning, we solve the LP in Equation 3 to repair the network; (2) **PRoViT_FT** (abbrev. **FT**): we run fine tuning on the last layer until the repair set is satisfied; (3) **PRoViT_FT+LP** (abbrev. **FT+LP**): we run Algorithm 1, which alternates between PRoViT_FT and PRoViT_FT. We compare the performance of these approaches by evaluating their efficiency, drawdown, and generalization on various repair specifications.

***Repair Set.*** We repair weather-corrupted images from the ImageNet-C dataset (Hendrycks & Dietterich, 2018). We select $|K|$ random ImageNet labels and repair 500 images for each of the selected labels. The 500 images are selected by choosing **4** corruptions of **5** base images: fog, brightness, frost, and snow. Figure 1 shows example images with the four corruptions. Each corruption has **5** severity levels. We apply **5** rotations ($-10°, -5°, 0°, 5°, 10°$) to each image. In total, this creates $4 \times 5 \times 5 = 500$ images per label in the repair set. In this experiment, we increase the size of the repair set by incrementing the number of labels we include. Thus, the size of each repair set is $|K| \times 500$.

***Drawdown Set.*** We use the official ILSVRC2012 ImageNet validation set (Deng et al., 2012) to measure the drawdown of each repair. For ViT-L/32, the top-1 accuracy is 76.972% and the top-5 accuracy is 93.07%. For DeiT, the top-1 accuracy is 81.742% and the top-5 accuracy is 95.586%. For ResNet152, the top-1 accuracy is 78.312% and the top-5 accuracy is 94.046%. For VGG19, the top-1 accuracy is 72.376% and the top-5 accuracy is 90.876%.

***Generalization Set.*** The generalization set includes all of the weather-corrupted ImageNet-C images within the selected $|K|$ labels that are not present in the repair set. There are **4** corruptions of the remaining **45** base images. Each corruption has **5** severity levels and we apply the same **5** rotations to each image. So each label, there are $4 \times 45 \times 5 \times 5 = 4500$ images in the generalization set. In total, the size of each generalization set is $|K| \times 4500$ where $|K|$ is the number of labels in the repair set.

### 4.1 COMPARISON WITH BASELINE

We compare the performance of our approach, PRoViT$_{FT+LP}$, against the baseline, FT$_{all}$. Table 1 shows the drawdown and generalization for each Vision Transformer. FT$_{all}$ results in terrible drawdown, causing the Vision Transformer to lost most of its original test set accuracy. In addition, PRoViT$_{FT+LP}$ consistently outperforms FT$_{all}$'s generalization by 20%. PRoViT$_{FT+LP}$ also results in near-zero drawdown, never reaching more than 1%. Table 2 shows the time comparison between our baseline FT$_{all}$ and our approach PRoViT$_{FT+LP}$. FT$_{all}$ takes significantly more time to repair than PRoViT$_{FT+LP}$, so we set a timeout of 20000 seconds. Even as repair set sizes reach into the thousands, PRoViT$_{FT+LP}$ is more efficient than FT$_{all}$ was on much smaller repair sets.

### 4.2 ABLATION STUDY

We compare the variations of PRoViT, as shown in both Table 1 and Table 2. For ViT-L/32, both the FT and LP variants of PRoViT$_{FT+LP}$ are faster than FT+LP. However, for DeiT, the LP variant is always the fastest. The PRoViT$_{LP}$ always achieves the best drawdown. The drawdown is sometimes even negative, meaning that instead of "forgetting" prior knowledge, additional accuracy was gained on the test set. PRoViT$_{FT}$ has the best overall generalization.

The results for the FT+LP variation of PRoViT provide evidence of a nice tradeoff between drawdown and generalization. FT+LP leverages the benefits of both variations while still providing provable correctness guarantees.

These results demonstrate PRoViT's efficiency, low drawdown, and high generalization, all while maintaining correctness guarantees on the repair set. PRoViT's success highlights the strength of targeting the last layer of a Vision Transformer for repair.

### 4.3 COMPARISON ACROSS ARCHITECTURES

To further explore the uniqueness of PRoViT's approach to repairing Vision Transformers, we conducted experiments on ResNet152 and VGG19 to demonstrate that PRoViT is not well suited for other image recognition architectures. Table 3 shows the drawdown and generalization of the different variants of PRoViT. The generalization is good, but the drawdown is extremely high for both ResNet152 and VGG19. Notably, the FT approach resulted in the best drawdown, but the repairs are considerably worse than those on the Vision Transformers. This provides insight into the key differences between the ways convolutional architectures like ResNet and VGG distill information within images and how that information is reflected in the final output layer. The theoretical basis for why PRoViT works well on Vision Transformers but not convolutional networks is left to future work.

## 5 RELATED WORK

### 5.1 FORMAL METHODS FOR TRAINING AND VERIFICATION

Training DNNs that are robust to adversarial inputs has been extensively researched. Müller et al. (2023) present a certified DNN training approach that evaluates worst-case loss on small boxes within the adversarial input region. Balunovic & Vechev (2019) propose adversarial training in combination with provable defenses to achieve certified robustness. Their approach aims to strike a balance between high test accuracy and providing robustness certificates.

There are also formal verification methods that work to prove the correctness of a particular input according to some specification for a pre-trained DNN. DeepPoly (Singh et al., 2019) is a verification tool that uses abstract transformers to prove properties of DNNs. DeepT (Bonaert et al., 2021) is also a verification tool based on abstract interpretation, specific to Transformer architectures. Both DeepPoly and DeepT return counterexamples to the properties if they are found. These counterexamples can be used as input to PRoViT. Shi et al. (2020) have also addressed the verification of Transformers by computing certified bounds to reflect the importance of specified inputs. Their experiments, along with those in DeepT, focus mainly on NLP Transformers as opposed to Vision Transformers.

## 5.2 PROVABLE REPAIR OF DNNS

The provable repair problem is a related but separate problem for DNNs. Certified training operates as a starting point for correctness guarantees, usually creating a model from scratch. Verification methods aim to produce a certificate of correctness on a pre-trained model; it does not make edits to the DNN at all. Provable repair, on the other hand, provides correctness guarantees for specified inputs by editing the model's parameters.

There are two classes of provable repair approaches for DNNs: architecture-modifying and architecture-preserving. The first architecture-modifying approach proposed is PRDNN (Sotoudeh & Thakur, 2021), which processes a DNN by decoupling its structure. This architecture modification allows PRDNN to provide correctness guarantees about the parameter edits by formulating the problem as an LP. REASSURE (Fu & Li, 2022) is another architecture-modifying provable repair approach. REASSURE adds small "patch networks" to the original DNN architecture that activate for the inputs that are in the repair set. The parameters of the patch networks can be designed to correct the behavior of the designated points in the repair set. REASSURE does not work on Vision Transformer architectures due to the nonlinear activation functions within the encoder layers.

Architecture-preserving provable repair methods guarantee the correctness of the inputs post-repair without modifying the original structure of the DNN. Goldberger et al. (2020) first proposed formulating the repair as an LP in their approach "minimal modifications of deep neural networks" (MMDNN), but due to the nonlinear nature of the activation functions, the method only works for large DNNs when the repair is restricted to the last layer. Otherwise, the search space of potential LP solutions becomes exponential and does not scale. APRNN (Tao et al., 2023) also formulates the repair problem as an LP, but avoids the exponential search space of MMDNN by adding activation pattern constraints to the LP. Thus, APRNN can successfully repair any layer of a DNN. However, on such large models like Vision Transformers, neither MMDNN nor APRNN scale, even when restricted to just the last layer. PRoViT is hence the only method architecture-preserving provable repair method that scales to Vision Transformers.

## 5.3 TRANSFORMER EDITING

While none of the prior provable repair approaches have focused on Transformer architectures, there are many approaches in recent research that focus on editing Transformers without formal correctness guarantees. SERAC (Mitchell et al., 2022) tackles the Transformer editing problem by storing edits in an explicit memory, acting as a wrapper around the base Transformer model. In addition to the memory-based cache, SERAC also trains a scope classifier and counterfactual model to determine when to override the base model during inference. Transformer-Patcher (Huang et al., 2023) is another approach that, similarly to PRoViT, makes edits to the last layer of a Transformer. For each input to correct, however, Transformer-Patcher adds a neuron to the last layer to fix the output's behavior. This approach suffers from scalability issues and increases the inference time of the resulting Transformer model significantly.

ROME (Meng et al., 2022) is another model editing approach for Transformers based on identifying neuron activations that determine a model's predictions. The weights of a Transformer are updated based on these selected neurons to correct a particular "fact" in an NLP Transformer. ROME only has the capability to update one fact at a time, so its scalability is restricted. MEMIT (Meng et al., 2023) builds on ROME by tracing a "critical path" through the MLP layers and updates the weights along this critical path to allow for thousands of edits at once. This addresses the scalability issue of ROME, however both MEMIT and ROME require the NLP facts to be in the form of a (subject, relation, object) format, and thus are not flexible to other types of model edits.

## 6 CONCLUSION

We presented PRoViT, a scalable architecture-preserving provable repair approach for Vision Transformers. We leveraged the combination of fine tuning and linear programming to make edits to the last layer of Vision Transformers. Our experimental evaluation demonstrates that PRoViT is efficient, generalizes well, and avoids drawdown, all while providing provable correctness guarantees on the repair set. We highlight that for provable repair of Vision Transformers, the last layer is all you need.

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
