

Figure 2: Representation of the number of labels that can be included in the repair set for ViT-L/32 with 100GB of memory. The shaded regions demonstrate the possible values of $N_K$ for a given repair set size. The upper bounds estimate the value of $N_K^\uparrow$ (y-axis) given a desired repair set size $N_\mathcal{S}$ (x-axis). In this setting, $N_K^{\max} = 395$, $N_K^* = 69$, $N_\mathcal{S}^{\max} = 8598$, and $N_\mathcal{S}^* = 2765$.

## A  THE MAXIMUM NUMBER OF LABELS THAT CAN BE INCLUDED IN A REPAIR SET

This section addresses the question of how many labels in a repair set PRoViT can handle. The number of labels included in the repair set is a bottleneck due to the memory needed to solve the linear program within PRoViT. As shown in Remark 2, the number of variables $N_v = p \times |K| + |K|$ and the number of constraints $N_c = |\mathcal{S}| \times |K|$, where $|\mathcal{S}|$ is the size of the repair set and $|K|$ is the number of unique labels in the repair set. Given a Vision Transformer to repair, $p$ is fixed. For example, $p = 768$ for ViT-L/32. Thus, the capacity of the repair on a given Vision Transformer depends entirely on $K$ and $\mathcal{S}$. For ease of notation, we use $N_\mathcal{S}$ to denote the size of the repair set $|\mathcal{S}|$ and $N_K$ to denote the number of labels in the repair set $|K|$.

In general, estimating how much time and memory solving the LP will take is difficult due to the many heuristics employed by modern LP solvers such as Gurobi (Gurobi Optimization, LLC, 2023). We know that $N_K < 1000$ for ImageNet Vision Transformers because if $N_K = 1000$, PRoViT$_{\text{LP}}$ becomes essentially equivalent to APRNN (Tao et al., 2023) and MMDNN (Goldberger et al., 2020), and those approaches do not scale.

In this section, we provide some empirical data and methods to estimate the maximum number of labels that can be included in a repair set ($N_K$) given the size of the repair set ($N_\mathcal{S}$). To evaluate the capacity of repairs for maximum $N_K$ and $N_\mathcal{S}$ values, we performed binary searches on repair sets with those values. We conducted this experiment on the Vision Transformer ViT-L/32 with a memory limit of 100GB, which is a smaller memory limit than what was used in Section 4 but allowed us to run the search on multiple machines with similar hardware constraints.

For the Vision Transformer ViT-L/32 and a memory limit of 100GB, one can estimate the maximum number of labels given a repair set size by referring to the shaded regions in Figure 2. The x-axis represents the size of the repair set, $N_\mathcal{S}$. The y-axis represents the number of labels that can be included in the repair set, $N_K$. The upper bounds of the figure represent the maximum possible number of labels that can be included in the repair set, $N_K^\uparrow$.

We describe the process to obtain this figure to estimate $N_K^\uparrow$ below:

**Definition 4.** *Let $N_K^{max}$ be the maximum number of labels in the repair set possible given a Vision Transformer $\mathcal{N}$ and memory limit $M$.*

To find $N_K^{\mathrm{max}}$, $N_S$ must equal $N_K$. In other words, we can only include one image per label in the repair in order to maximize the number of labels, $N_K^{\mathrm{max}}$. For ViT-L/32 with 100GB of memory, $N_K^{\mathrm{max}} = 395$.

**Definition 5.** *Let $N_S^{max}$ be the maximum size of the repair set possible given a Vision Transformer $\mathcal{N}$ and memory limit $M$.*

To find $N_S^{\mathrm{max}}$, $N_K$ must equal 1. In other words, if we want to include as many images in the repair set as possible, we should not exhaust available memory resources by including more labels. For ViT-L/32 with 100GB of memory, $N_S^{\mathrm{max}} = 8598$.

**Definition 6.** *Let $N_S^*$ be the maximum repair set size possible with respect to the $N_K^{max}$ value defined above. Given a $N_K^{max}$ for a Vision Transformer $\mathcal{N}$ and memory limit $M$, $N_S^*$ is the maximum number of images that can be included in the repair set.*

$N_S^* \ll N_S^{\mathrm{max}}$. For ViT-L/32 with 100GB of memory, $N_S^* = 2765$.

**Definition 7.** *Let $N_K^*$ be the maximum number of labels in the repair set possible with respect to the $N_S^{max}$ value defined above. Given an $N_S^{max}$ for a Vision Transformer $\mathcal{N}$ and memory limit $M$, $N_K^*$ is the maximum number of labels that can be included in the repair set.*

$N_K^* < N_K^{\mathrm{max}}$. For ViT-L/32 with 100GB of memory, $N_K^* = 69$.

We now describe how to construct and interpret Figure 2 using the above definitions. The green region is constructed based on $N_K^{\mathrm{max}}$, which we know to be 395 in this experimental setting. Because it is impossible for the number of labels to be greater than the size of the repair set, we know that $N_K < N_S$. Thus, the line $N_S = N_K$ serves as our upper bound for the green region ($1 \leq N_S \leq 395$).

We construct the red region based on $N_S^*$, which is 2765 in this scenario. We know that it is impossible to have more than 395 labels in the repair set. $N_S^*$ is the maximum repair set size with 395 labels. Thus, the line $N_K = 395$ is the upper bound for the red region ($395 < N_S \leq 2765$).

Finally, we construct the blue region via a linear approximation with the endpoints $(N_S^*, N_K^{\mathrm{max}})$ and $(N_S^{\mathrm{max}}, N_K^*)$ due to the inherent relationships between these values as defined above. In this experimental setting, these values are (2765, 395) and (8598, 69) respectively. The resulting linear approximation is $-0.0559N_S + 549.5$, which serves as the upper bound for the blue region ($2765 < N_S \leq 8598$).

Given a desired repair set size $N_S$, one can refer to Figure 2 to determine the maximum number of labels possible, $N_K^\uparrow$. However, any $(N_S, N_K)$ pair in the shaded regions serves as a reliable estimation of $N_K$ for a given repair set size $N_S$ on ViT-L/32 with 100GB of memory.

## B    BASELINE COMPARISON WITH APRNN

The following experiment contains a baseline comparison with APRNN (Tao et al., 2023), a state-of-the-art LP-based provable repair approach. Because APRNN does not scale to the experiments included in Section 4, we have reduced the Vision Transformers to only include the first 50 ImageNet classes instead of the full 1000.

The experimental setup is as follows:

***Repair Set.***    We repair weather-corrupted images from the ImageNet-C dataset (Hendrycks & Dietterich, 2018). We select the first $|K|$ ImageNet labels and repair 100 images for each of the selected labels. The 100 images are selected by choosing **4** corruptions of **5** base images: fog, brightness, frost, and snow. Each corruption has **5** severity levels. In total, this creates $4 \times 5 \times 5 = 100$ images per label in the repair set. In this experiment, we increase the size of the repair set by incrementing the number of labels we include. Thus, the size of each repair set is $|K| \times 500$.

***Drawdown Set.***    We use the official ILSVRC2012 ImageNet validation set (Deng et al., 2012) to measure the drawdown of each repair.

***Generalization Set.***    The generalization set includes all of the weather-corrupted ImageNet-C images within the selected $|K|$ labels that are not present in the repair set. There are **4** corruptions of

Table 4: Drawdown and generalization results in the reduced Vision Transformer experiment with just 50 ImageNet classes. **Bold number** indicates the *best result*. Recall that lower drawdown is better and higher generalization is better.

| Model | $|K|$ | $|S|$ | Drawdown [%] | | | | Generalization [%] | | | |
| --- | --- | --- | --- | --- | --- | --- | --- | --- | --- | --- |
| | | | APRNN | PRoViT | | | APRNN | PRoViT | | |
| | | | | LP | FT | FT+LP | | LP | FT | FT+LP |
| ViT-L/32 | 4 | 400 | **-4.50%** | **-4.50%** | **-4.50%** | **-4.50%** | **1.80%** | 1.48% | 1.09% | 1.61% |
| | 8 | 800 | -5.00% | -5.00% | -5.00% | **-5.25%** | 8.20% | 8.15% | **10.47%** | 8.50% |
| | 12 | 1200 | -3.67% | **-3.83%** | 5.67% | -1.00% | **7.98%** | 7.78% | 4.84% | 6.26% |
| | 16 | 1600 | **-3.38%** | -3.25% | -2.38% | **-3.38%** | 5.95% | 5.98% | 4.86% | **6.14%** |
| | 20 | 2000 | -2.80% | **-3.20%** | 0.80% | -0.90% | 7.73% | **7.95%** | 4.03% | 5.93% |
| DeiT | 4 | 400 | **-2.00%** | **-2.00%** | **-2.00%** | **-2.00%** | **0.60%** | **0.60%** | -0.36% | **0.60%** |
| | 8 | 800 | **-2.00%** | -1.75% | -1.50% | -1.75% | 3.35% | 4.48% | **6.85%** | 5.42% |
| | 12 | 1200 | **2.00%** | 2.17% | 8.17% | 3.83% | 10.42% | **11.20%** | 7.91% | 9.68% |
| | 16 | 1600 | -1.00% | **-1.37%** | 0.00% | -0.87% | 6.99% | 7.23% | 3.58% | **7.89%** |
| | 20 | 2000 | **-0.40%** | **-0.40%** | 3.90% | 1.20% | **10.83%** | 10.71% | 6.36% | 8.85% |

Table 5: Time spent in the reduced Vision Transformer experiment with just 50 ImageNet classes. **Bold number** indicates the *best result*.

| Model | $|K|$ | $|S|$ | Time [s] | | | |
| --- | --- | --- | --- | --- | --- | --- |
| | | | APRNN | PRoViT | | |
| | | | | LP | FT | FT+LP |
| ViT-L/32 | 4 | 400 | 283s | 80s | 217s | **79s** |
| | 8 | 800 | 1520s | **169s** | 418s | 170s |
| | 12 | 1200 | 1983s | **246s** | 614s | **246s** |
| | 16 | 1600 | 3309s | 346s | 1510s | **345s** |
| | 20 | 2000 | 3360s | **468s** | 1881s | 470s |
| DeiT | 4 | 400 | 244s | **92s** | 228s | **92s** |
| | 8 | 800 | 383s | 186s | 440s | **185s** |
| | 12 | 1200 | 1761s | 283s | 652s | **282s** |
| | 16 | 1600 | 1746s | 387s | 1608s | **386s** |
| | 20 | 2000 | 1941s | 506s | 1998s | **478s** |

the remaining **45** base images. Each corruption has **5** severity levels. So for each label, there are $4 \times 45 \times 5 = 900$ images in the generalization set. In total, the size of each generalization set is $|K| \times 900$ where $|K|$ is the number of labels in the repair set.

***Results.*** Table 4 shows that APRNN and PRoViT perform similarly on both drawdown and generalization. Neither method outperforms the other by a significant margin. However, Table 5 shows that PRoViT significantly outperforms APRNN on runtime. In this experiment, we observe speedups between 2x and 9x when comparing PRoViT to APRNN.