# OpenReview forum: "Provable Repair of Vision Transformers: Last Layer is All You Need"
_ICLR.cc/2024/Conference — Submitted to ICLR 2024_

### Official Review · Reviewer_QPLU · 2023-10-27

**Soundness:** 2 fair
**Presentation:** 3 good
**Contribution:** 3 good
**Rating:** 5
**Confidence:** 4

**Summary:**

This paper talks about on how to add robustness in VIT models, something which was guaranteed in VGG and ResNet.
They achieved this using a set patter of modifying last layers of VIT model.

**Strengths:**

* Great theoretical backing for their proof.
* Mentions initial hypothesis what they are trying to achieve using this task. This is a great way to start any research problem.

**Weaknesses:**

* I fail to see lot of practical sense from this paper.
* Adding some pictorial references to the idea might be able to help readers to grasp the idea completely.

**Questions:**

NA

---

> ### Author Response · Authors · 2023-11-18
>
> Thanks so much for taking the time to review our paper!
>
> We have added some figures representing the types of ImageNet images we repair in our experimental evaluation.
>
> In terms of clarifying the practical sense of the paper, we believe that provable repair of Vision Transformers is incredibly important for scenarios in which Vision Transformers are used in safety-critical domains such as self-driving automobiles and medical diagnostics. We have cited additional references in the introduction that justify the necessity for provable repair of Vision Transformers.
>
> Thank you for the suggestions!

---

### Official Review · Reviewer_ub3F · 2023-11-06

**Soundness:** 3 good
**Presentation:** 3 good
**Contribution:** 2 fair
**Rating:** 5
**Confidence:** 4

**Summary:**

The paper presents a method for provable repair the correction of classification outputs using Vision Transformers on a specified set of images with certain guarantees and with limited degradation of performance (drawdown) on the initial training set. The proposed provable repair method claims that suitable modification of the final fully-connected layer of a vision transformer is sufficient to achieve these goals. Specifically, the paper builds on provable repair methods proposed for DNNs, like MMDNN and APRNN, that, while they cannot be fully implemented in the case of vision transformers (ViTs) due to the presence of self-similarity modules, they can be applied to the last fully-connected layer of ViTs.

**Strengths:**

The paper is well written and easy to read. The main ideas behind the proposed method are clearly exposed and their relation to previous work is presented. Based on concepts introduced in the MMDNN and ARPNN works, a modification of the last layers weights is performed by defining a linear programming problem for satisfying the required constrains. To improve the efficiency of the method, the problem is limited only to the class labels contained in the given repair set. Additionally, fine-tuning of the final layer is also applied in an alternating way, to further improve generalization and limit drawdown.

**Weaknesses:**

The contribution is somewhat limited, as the main idea is based on similar approaches applied for DNNs, such as MMDNN (Goldberger et al., 2020) and APRNN (Tao et al., 2023). Nevertheless, these methods have been suitably modified for their application in the context of ViTs.

Some aspects of the evaluation could be improved. For example, it is mentioned that standard LP-based repair of the last layer could not be considered due to scalability issues. While this is a major limitation of this baseline, it would be interesting to provide some results on a dataset with a reduced number of categories that would make this comparison possible. Also, another aspect that would be interesting to consider in the ablative study is the performance of the proposed method for large values of K. Is there a maximum K value in practice and, if yes, how can this limit be estimated in practice?

**Questions:**

As stated above, is there a maximum K value inpractice? If yes, is it easy to estimate it?

---

> ### Author Response · Authors · 2023-11-18
>
> Thanks so much for taking the time to review our paper!
>
> We are running an experiment with a reduced number of ImageNet categories so that we can compare PRoViT against APRNN (Tao et al., 2023). We will include the results of this experiment in the Appendix.
>
> We will also include an experiment that tests the limits of the maximum K value and determine whether this value is easy to estimate in practice. Thank you for the suggestions on additional experiments to include!

---

> > ### Comment · Reviewer_ub3F · 2023-11-22
> > **Response to authors**
> >
> > I thank the authors for their feedback. I think covering these topics will make their work stronger.

---

> > > ### Author Response · Authors · 2023-11-23
> > >
> > > We have just uploaded the results of the experiments that you suggested.
> > >
> > > The first experiment compares the performance of APRNN (Tao et al., 2023) with PRoViT on Vision Transformers with a reduced number of ImageNet categories. After reducing the Vision Transformers to just 50 ImageNet categories, we observed that PRoViT and APRNN achieve similar drawdown and generalization, but PRoViT significantly outperforms APRNN in runtime.
> > >
> > > To answer the question of finding the maximum K value, we conducted a conceptual and empirical study. By setting a specific memory limit and Vision Transformer to repair, we explored the limits of the repair set size and the number of labels in the repair set.
> > >
> > > The results of both experiments are in the Appendix (Supplementary Material). Thank you again for your suggestions for additional experiments.

---

### Official Review · Reviewer_CCeY · 2023-11-11

**Soundness:** 1 poor
**Presentation:** 2 fair
**Contribution:** 2 fair
**Rating:** 3
**Confidence:** 4

**Summary:**

This paper proposes a method for provable repair of neural networks by modifying only the weights of the last linear layers so as to maximise the classification accuracy on the repair set. This problem can be cast as a linear program (LP), as previously done in the literature. The novelty of this work that sparsity is taken into account when solving this LP.

**Strengths:**

The proposed method can exploit the sparsity in the LP (given that the number of labels in the repair set remains small). The results on ViT are quite good.

**Weaknesses:**

1. The authors claim that their method has "provable correctness guarantees" (all images in the repair set will be classified correctly post-repair), which is of course not true because the final accuracy deepens on the repair set and the capacity of the last layer. (Take the ImageNet training set as the repair set for example, can the method guarantee a 100% accuracy? Obviously not.) This incorrect claim seems to come from a misconception regarding the algorithm, as discussed below in the next point.

2. There's a fundamental algorithmic issue. "Since FTall only terminates once all inputs are classified correctly, it is a provable repair approach." (at the end of section 2), and "If there is no solution to the LP, the loop continues. Otherwise, the repaired Vision Transformer is returned." (at the end of page 5): the authors claim that their algorithm terminates when all the images are correctly classified. The question is: Is it guarantee to terminate?

3. Some other claims are also questionable. For example, "PRoViT scales to thousands of images": this depends on the number of classes present in the repair set. For example, if there's only a single class, then this claim becomes uninteresting.

4. Are you sure that the optimization problems in Equations (2) and (3) are linear programs? In addition, please avoid using "Theorem" for everything. For example, Theorems 2 and 4 should be stated as remarks, in my opinion. (And perhaps the other ones should be labeled with Proposition instead of Theorem.)

5. Most importantly, the proposed method has nothing to do with vision transformers. It is largely based on existing works, the optimization problem is the same, only the resolution has been improved by taking into account sparsity. And it also works for any models other than ViT (as long as they has a linear layer at the end, which is the case for all common architectures). It happens to work well for ViTs, but there was absolutely no explanation for this phenomenon, and ViTs were also not the motivation for the design of the method, so the title and the framing of this paper seem to use ViTs only to attract attention.

**Questions:**

See weaknesses.

---

> ### Author Response · Authors · 2023-11-18
>
> Thanks for taking the time to review our paper! We will address your comments in the following ways:
>
> 1. In our presentation of the LPs in Sections 2 and 3, we originally noted that an LP solver can find a solution “if it exists.” However, we also included such language in the introduction to be clear about the soundness vs. completeness, in line with prior provable repair approaches. Our approach is sound; that is, if the approach says that it has repaired the network, then the resulting network is guaranteed to satisfy the repair specification. Similar to all prior provable repair approaches, our approach is not complete; that is, it may not be able to find a network that satisfies the given repair specification. The reason for such incompleteness could either be because such a network does not exist due to, for instance, a contradictory repair specification or because the approach is restricted to modify only certain parameters of the network. (Note that Section 5 of APRNN (Tao et al., 2023) shows that the algorithm of REASSURE (Fu and Li, 2022) is not sound or complete.)
> 2. The issue of termination is merely related to how one expresses an approach that is sound but not complete. One alternative to describing such an approach is for the algorithm to always terminate and return either a solution or $\bot$ (representing “I could not find a solution”). In this case, one would prove that if the return value is not $\bot$, then the answer is correct (sound). A second alternative is to only terminate when a solution has been found. In this second case, one would prove that any value returned by the algorithm is correct. However, the algorithm might not terminate.  We just happened to pick this latter approach for describing $FT_{all}$ in Section 2 and $PRoViT_{LP+FT}$ in Algorithm 1.
> &nbsp;&nbsp;&nbsp;&nbsp;In practice, we use a timeout to terminate the algorithm (both $FT_{all}$ and $PRoViT_{LP+FT}$), so in practice we have implemented the first alternative described above. Furthermore, in our experiments $FT_{all}$ timed out while PRoViT always terminated within the timeout limit.
> &nbsp;&nbsp;&nbsp;&nbsp;We have also clarified the presentation of $FT_{all}$ at the end of Section 2. $FT_{all}$ is a baseline, not part of PRoViT. $FT_{all}$ is just regular fine tuning across the entire Vision Transformer with the termination condition of achieving 100% accuracy on the repair set.
>
> 3. None of the prior provable repair approaches are able to repair thousands of images for ImageNet Vision Transformers with low drawdown and high generalization, even when restricted to just one class. APRNN runs out of memory, fine tuning times out or causes high drawdown, etc. Hence, we believe that the claim that PRoViT can repair thousands of images is interesting. For example, it would be incredibly useful to repair thousands of instances of a single class such as a “stop sign” for an autonomous vehicle.
> 4. Yes, we are certain that Equations 2 and 3 are LPs. Is there notation that we have used that is confusing in our LP formulation? Please let us know so we can rectify that. We have changed the Theorems to Remarks and Propositions.
> 5. This work was originally inspired by trying to find a provable repair method for Vision Transformers, because one does not already exist. We were pleasantly surprised to find that by only focusing on repairing the last layer, we were able to find a simple, efficient, and effective way to repair Vision Transformers. The novelty of this work is not only taking sparsity into account in the LP formulation of the repair, but also that the combination of fine tuning and LP-based repair of the last layer strikes a balance between drawdown and generalization. Our experiments on ResNet and VGG (Section 4) show that focusing on the last layer does not work as well as it does on Vision Transformers, so we still believe that this approach is ViT-specific. Our hypothesis for this phenomenon is that because the last layer of the Vision Transformer is an output vector containing the “class token” from the final encoder layer, the class information encoded in that vector is more robust than the information that is just the result of convolutions. However, further theoretical justification for this hypothesis is left to future work. We have added this note regarding future work in Section 4.
>
> Thank you for the suggestions!

---

### Meta-Review · Area_Chair_mZL6 · 2023-12-30

**Metareview:**

The paper attempts to improve repair of deep neural networks on several axes: scalability, drawdown, generalization, etc. We agree with authors that this is "an important and growing topic in trustworthy machine learning". The paper presents a method by modifying only the final classification layer with a combination of fine-tuning and linear programming. Empirically the proposed method works better for repairing Vision Transformers (ViTs) than other architectures. The paper received a lukewarm response from the reviewers who raised several concerns including limited novelty, limited evaluation, overstated claims, etc. We thank both the authors and reviewers for engaging during the discussion period towards improving the paper and providing new experiments. The author response helped resolve some concerns like limited evaluation but highlighted other limitations of the proposed work like other baselines matches in terms of drawdown and generalization. Although scalability is still superior, given other shortcomings in the current draft, it would require different framing of the story and more experiments which require another round of reviewing before the paper can be accepted.

**Justification For Why Not Higher Score:**

Limited novelty: The core concept largely builds on existing methods.
Overstated claims: Some claims about provable correctness and scalability may be overstated or misleading. There are concerns about the algorithm's ability to terminate.
Limited practical application: Limited to one architecture empirically and no discussion on why it doesn't work on other architectures? Is it because termination condition is not easily satisfied by embeddings from other architectures? Another limitation is that it is assumed |K| << |S|, if this is not the case then proposed method doesn't offer much benefits.

**Justification For Why Not Lower Score:**

N/A

---

### Decision · Program_Chairs · 2024-01-16

Reject